

# Inverting Rayleigh surface wave velocities for eastern Tibet and western Yangtze craton crustal thickness based on deep learning neural networks

Xian-Qiong Cheng[1] , Qi-He Liu[2], Ping-Ping Li[1]

[1] College of Geophysics, Chengdu University of Technology, Chengdu , P.R. China
[2] The School of Computer Science and Engineering, University of Electronic Science and Technology of China, Chengdu , P.R. China

*Correspondence to:* Xian-Qiong Cheng (chxq@cdut.edu.cn)

**Abstract.** Crustal thickness is an important factor affecting lithosphere structure and therefore deep geodynamics. In this paper, we propose to apply deep learning neural networks called stacked sparse auto-encoder to obtain crustal thickness for eastern Tibet and western Yangtze craton. Firstly taking phase and group velocities simultaneously as input and theoretical crustal thickness as output, we construct twelve deep neural networks trained by 70,000 and tested by 30,000 theoretical models. We then invert observed phase and group velocities by these twelve neural networks. Based on test errors and misfits with other crustal thickness models, we select the optimized one as crustal thickness for study areas. Compared with other ways detected crustal thickness such as seismic wave reflection and receiver function, we conclude that deep learning neural network is a promising, believable and inexpensive tool for geophysical inversion.

## 1 Introduction

Tibetan plateau is an example of a large orogenic plateau formed as a result of Euro-Asian continent and Indian continent collision. The morphology of the region along the eastern margin of the Tibetan plateau, adjacent to the strong rigid crustal basement of the Sichuan basin, is characterized by very steep relief with high mountain ranges and steep peaks (Clark et al., 2004; Burchfiel et al., 1995; Zhu et al.,2012). Longmen mountain fault occurred Wenchuan earthquake of 12 May 2008 and Lushan earthquake of 20 April 2013 is between  Tibetan plateau and the Sichuan basin. In this paper, we try to attain crust thickness for eastern Tibet and western Yangtze craton and analysis geodynamic implications. As we all know, the more we know the characteristic and composition of crust which is an important part of lithosphere, the further we investigate deep earth. Discontinuity between crust and mantle called moho discontinuity is an important one for geodynamics such as crustal evolution, tectonic activities and so on, in addition to the correcting gravity for the crustal effects, seismic tomography and geothermal modeling.  The depth of moho or called crust thickness varies greatly over small length scales and has significant effects on fundamental mode surface waves(Ueli Meier et al.,2007).There are several methods to get moho depth, such as deep seismic sounding profile for china continent(Zeng et al.,1995), inverting satellite gravity data to get whole global crust and lithophere thickness(Fang et al.,1999), inverting Bouguer gravity and topography data to get moho depth for china and its  adjants (Huang et al.,2008; Guo et al., 2012),inverting receiver function to get moho depth and Possion's ratio for china continent (Chen et al.,2010;Zhu,2012). Especially, a newest crust model called crust1.0 at $1^{o} \times 1^{o}$ (Laske et al.,2013; Stolk, et al., 2013) are based on refraction and reflection seismology as well as receiver function studies. As a consequence, resolution and consistence among different crust models are high in regions with good data coverage and uncomplicated structure but in regions with poor or no data coverage or complicated structure crustal thickness estimates are largely extrapolated. In order to overcome these defaults, another kind of fully non-linear method called neural network to put forward to get crustal thickness(Devile et al.,1999;Ueli Meier et al.,2007).

Dispersion characteristic of surface wave provide a powerful tool to research structure of crust and upper mantle. So far phase and group velocity measurements of fundamental mode surface waves are most commonly used to constrain shear-velocity structure in the crust and upper mantle on a global scale (Zhou et al. 2006) or on regional scale (Zhu et al.,2002), while a few works to invert fundamental mode surface wave data for global or regional crustal thickness and to present a global or regional crustal thickness model(Devile et al.,1999; Ueli Meier et al.,2007; Das & Nolet 2001; Shapiro & Ritzwoller ,2002). In this article, we will investigate how to retrieve the crustal thickness for eastern



Tibet and western Yangtze craton from newest and high-resolution phase and group velocity maps (Xie et al.,2013). As seismology points out that there are many factors affect phase and group velocity, and inverting them for discontinuities within the earth forms a non-linear inverse problem(Ueli Meier et al.,2007). Because of strong non-linearity between crust thickness and surface wave dispersion and large variance of crust thickness we cannot treat it with a linear inverse problem as Montagner & Jobert (1988) stated. As periods and method measured differently between group velocity and phase velocity, which the probing depths are different and measure error are largely independent, the simultaneous inversion of group velocity and phase velocity is substantially better than the use of either alone(Shapiro & Ritzwoller,2002). We focus on deep learning neural networks instead to solve the non-linear inverse problem, inverting crustal thickness from phase and group velocity measurements.

Since strong nonlinear relation among geophysical variables, neural networks have been widely used in different geophysical applications well summarized by van der Baan & Jutten (2000) such as in electrical impedance tomography(Lampinen &Vehtari ,2001), in seismic processing including trace editing, travel time picking, horizon tracking, and velocity analysis. Devilee et al.(1999) were the first to use a neural network to invert surface wave velocities for Eurasian crustal thickness in a fully non-linear and probabilistic manner. Ueli Meier et al.(2007) further develop the methods of Devilee et al. (1999), then invert surface wave data for global crustal thickness on a $2\circ \times 2\circ$ grid globally using a neural network. Although traditional shallow neural network can present nonlinear inverse function, it maybe cannot learn or approximate the real inverse function well when the real inverse function is too complicated. In contrast, deep learning neural network can overcome this problem since it has powerful representation abilities and can discover intricate structures in large data sets by using the back-propagation algorithm to indicate how a machine should change its internal parameters that are used to compute the representation in each layer from the representation in the previous layer (LeCun, Y.et.al.,2015).

To the best of our knowledge, we are the first to propose deep learning neural networks to learn and invert crustal thickness, which reveal crustal thickness is strong nonlinear with respect to phase and group velocity. The merits of our methods include: our method is inexpensive because we require a few observed data about phase and group velocities to obtain crustal thickness by using well-trained deep learning neural networks. Moreover, our deep learning neural networks train on vast synthetic models. Secondly, since deep learning neural networks can represent complex functions, it is possible to learn the crustal thickness inverse function precisely. Lastly, our results show changes of the number of neurons in each layer have little influence on test errors when the numbers of network layer achieve six and test errors are about 2.5e-6 , which indicates deep learning neural networks are robust to neural network structures with suitable layer numbers.

As Ueli Meier et al. (2007) demonstrated that the neural network approach for solving inverse problems is best summarized by three major steps: (1)forward problem. In this stage we proceed by randomly sampling the model space and solve the forward problem for all visited models. (2) designing a neural network structure. In this stage taking phase and group velocities as inputs and theoretical crustal thickness as outputs we train the deep learning neural networks and get an optimized one.(3) inverse problem. Base on trained networks we invert crustal thickness from observed phase and group velocities. In what follows we first give a short introduction to deep learning neural networks, and show how to train deep learning neural networks to model surface wave dispersion based on synthetic seismogram, then invert dispersion curves based on the trained networks. Finally we compare our crustal model with other crustal thickness models, and discuss the geodynamic implication implied by our model.

## 2   Deep  Learning Neural Networks

In geophysics the real inverse function is usually a complicated one with respect to geophysical observable variables. Traditional linear inversion modeling the real inverse function as a linear function can resolve linear relation problems however which depend on data coverage and initial models. Usually these linear methods can capture main information about the real inverse function. However, they cannot deal with nonlinear inverse functions. Neural network has its origins in attempts to find mathematical representations of information processing in biological systems(Bishop ,1995). The more deep strength of Artificial Neural Networks (ANNs) is, the more capabilities learn to infer complex, non-linear, underlying relationships without any a priori knowledge of the model(Bengio,2009). Traditional shallow neural network has gained in popularity in geophysics this last decade and has been applied successfully to a variety of problems such as well log,



interpretation of seismic data, geophysical inversion and so on. Although traditional shallow neural network can present nonlinear inverse function, it can only learn the relatively simple inverse function. In contrast, deep learning neural network has powerful representation ability and can apply a big geophysical observable data to learn and approximate the complicated inverse function well.

Based on the analysis above, we design deep learning neural network to obtain crustal thickness for eastern Tibet and western Yangtze craton. Compared with traditional shallow neural networks, deep learning neural network allows computational models that are composed of multiple processing layers to learn representations of data with multiple levels of abstraction and can learn complex functions. The essence of deep learning is building an artificial neural network with deep structures to simulate

the analysis and interpretation process of human brain for data such as image, speech, text, and so on. However, many research results suggest that gradient-based training of a deep neural network gets stuck in apparent local minima, which leads to poor results in practice. Fortunately, the greedy layer-wise training algorithm proposed by Hinton et.al 2006  to overcome the optimization difficulty of deep networks effectively. The training processing of deep neural networks is divided into two steps. Firstly,

unsupervised learning methods are employed to pre-train each layer parameters with the output of the previous layer as the input, giving rise to initialize parameter values. After that, the gradient-based method is used to finely tune the whole neural network parameter values with respect to a supervised learning criterion as usual. The advantage of the unsupervised pre-training method at each layer can help guide the parameters of that layer towards better regions in parameter space (Bengio,2009). There

are multiple types of deep learning neural network, such as convolutional neural networks, deep belief net and stacked Sparse Auto-encoders(sSAE). In this paper, we use sSAE to approximate the inverse function. The structure of sSAE is stacked by sparse autoencoders to extract abstract features. Here we introduce Sparse Auto-encoder briefly, and detailed description of the network training method is given by Liu.(2015).

A typical Sparse Auto-Encoder (SAE) can be seen as a neural network with three layers, as shown in Figure 1, including one input layer, one hidden layer, and one output layer. The input vector and the output vector are denoted by $v$ and $\hat{v}$, respectively. The matrix W is associated with the connection between the input layer and the hidden layer. Similarly, the matrix $\widehat{W}$ connects the hidden layer to the output layer. The vector $b$ and $\hat{b}$ are the bias vectors associated with the units in the hidden layer and

the output layer, respectively. The SAE is trained to encode the input vector $v$ into some representation so that the input can be reconstructed from that representation.  Let $f(x)$ denote the activation function, and the activation vector of the hidden layer then is calculated (with an encoder) as:

$$z=f(Wv+b), \qquad (1)$$

where z is the encoding result and some representation for the input v. The representation z, or code is

then mapped back (with a decoder) into a construction $\hat{v}$ of the same shape as v. The mapping happens through a similar transformation, e.g.:

$$\hat{v} = f(\widehat{W}z + \hat{b}) \qquad (2)$$

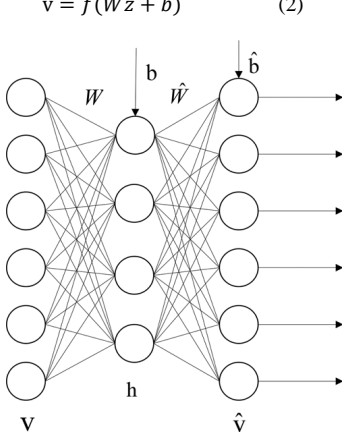

**Figure 1.** An auto-encoder with one hidden layer.(Liu et.al.,2015)

SAE is an unsupervised learning algorithm which sets the target values to be equal to the inputs and constrain output of hidden layer  which are near to zero and most hidden layer are inactive, the cost function is expressed as:



$$J_{sparse}(W, b) = J(W, b) + \beta \sum_{j=1}^{S_2} \rho \log \frac{\rho}{\hat{\rho}_j} + (1 - \rho) \log \frac{1 - \rho}{1 - \hat{\rho}_j} \quad (3)$$

Here $J(W, b)$ is cost function without sparsity constrain, $\beta$ controls the weight of the sparsity penalty term, $S_2$ is the number of neurons in the hidden layer, and the index j is summing over the hidden units in our network. $\hat{\rho}_j$ is the average activation of hidden unit j, $\rho$ is a sparsity parameter, typically a small
value close to zero.

Further, a stacked Sparse Auto-Encoder (sSAE) is a neural network consisting of multiple layers of SAE in which SAE are stacked to form a deep neural network by feeding the representation of the SAE found on the layer below as input to the current layer. Using unsupervised pre-training methods, each layer is trained as a sSAE by minimizing the error in reconstructing its input which is the output code
of the previous layer. After all layers are pre-trained, we add a logistic regression layer on top of the network, and then train the entire network by minimizing prediction error as we would train a traditional neural network. For example, a sSAE with two hidden layers is shown in Figure 2. This sSAE is composed of two SAEs. The first SAE consists of the input layer and the first hidden layer, and the representation or code of the input v is $h_1 = f(W_1 v + b_1)$. The second SAE comprises of two
hidden layers, and the code of $h_1$ is $h_2 = f(W_2 h_1 + b_2)$. Each SAE is added to a decoder layer as shown in Figure 1, and we can then employ unsupervised pre-training methods to train each SAE by expression (1). Finally, the matrix $W_1$, $W_2$, bias vector $b_1$ and $b_1$, are initialized. We then apply supervised fine-tuning methods to train entire network. Since our aim is calculating crustal thickness and this is a regression problem, we firstly attach a layer connected fully with last layer of the encoder
part (the matrix $W_s$). After that, we train this network as done in a traditional neural network.

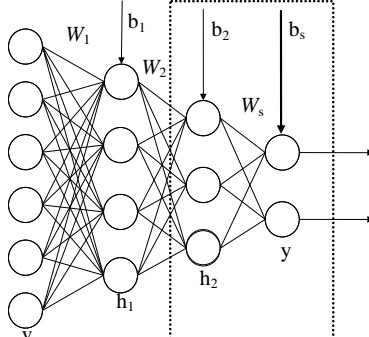

**Figure 2**: Stacked Sparse Auto-Encoder with two hidden layers.

### 3   Inverting surface wave data for crustal thickness

In this section, we'll introduce how to train a sSAE deep learning neural network and invert
crust thickness based on this trained network.

### 3.1 data preparation

We closely follow the model parametrisation methodology outlined in de Wit et al. (2014), which is based on PREM and is parametrised on a discrete set of 185 grid points used by Mineos. In addition, these models we've got show no correlations between physical parameters such as velocity, density, $\eta$
and attenuation profiles. As the model parametrisation methodology mentioned above, we generate 100,000 synthetic models based on the 1-D reference models PREM, which are randomly drawn from the prior model distribution, also prior ranges for the various parameters in our model are given in tables A.2–A.4.of de Wit et al.(2014). We use the Mineos package (Masters et al., 2014) to compute phase and group velocity for fundamental mode Rayleigh waves for all 100,000 synthetic 1-D earth
models.

As for observation data used in stage of inversion below, it is worth noting that in principle, group and phase velocities carry the same information, although group velocities are more sensitive to the shallow structure. Since a larger part of the signal is affected by the crustal structure, combination two types of data will constrain crustal thickness better in the presence of noise. The two are related by



$$U(T) = \frac{c(T)}{1 + \frac{T}{c(T)}\frac{dc(T)}{dT}} \qquad (4)$$

Where U denotes group velocity, C denotes phase velocity and T is period. Based on Rayleigh wave phase velocity from ambient noise(Xie et.al,2013), we compute corresponding group velocity according(4).

3.2 training sSAE deep learning neural network

As we all know, using a set of examples of corresponding input–output pairs, artificial neural networks can approximate an arbitrary non-linear function to solve the non-linear inverse problem. These examples are presented to a network in a so-called training process, during which the free parameters of a network are modified to approximate the function of interest(de Wit et al. 2014). Here adopting sSAE deep learning neural network, we take seismological observations (that is group and phase velocity of Rayleigh wave) as input, and get the output of earth structural parameters(that is crustal thickness).

Neural network training is sensitive to the random initialization of the network parameters. Therefore, it is common practice to train several neural networks with different initialisations, and subsequently choose the network which performs best on a given synthetic test data set, and the network which performed best on the test set is used to draw inferences from the observed data. After trying many times, we find the proportion of training data set to test one is 3:1 is reasonable. We've got final test errors which may be produced not only by different neural network structure decided by the number of inputting neuron, hidden layers and neuron in middle layer, also optional parameters such as number of train epochs and size of batch. What's more, type of activation function, value of learning rate, zero masked fraction, and value of non-sparsity penalty can affect final test errors. The table 1 below gives twelve cases and their corresponding test errors.

3.3 inverting crust thickness

Based on our all twelve neural networks, we invert Rayleigh phase velocities and group velocities (10~37.5mhz) to attain twelve crustal thickness models for eastern Tibet and western Yangtze craton. Considering not only the test errors of sSAE networks, also misfits and correlation coefficients of our twelve models with crustal thickness models from other research, we select network structure as shown in table 1 shown in ※. We find the best fit crustal thickness model from sSAE (Figure 3). We compare with same region crustal thickness from receiver function(Zhu J S et al.,2012), and two other global crustal thickness models, CRUST2.0 from Bassin et al. (2000) and the CUB2 model from Shapiro&Ritzwoller (2002)( Figure 4). The correlation coefficients of our model with ZJS, CRUST2.0 and CUB2 (Figure 5) are shown our model is best correlations with CUB2 and worst with ZJS because of model ZJS attained from receiver function has relatively sparse stations with poor data coverage and lower resolution.

Table1 deep learning neural network structures taking in this article

| sSAE Structure | parameters | | | | Error ×10⁻⁶ | CUB2 | | CRUST2.0 | | ZJS | |
|---|---|---|---|---|---|---|---|---|---|---|---|
| | Layers | D | E | F | ×10⁻⁶ | G | H | G | H | G | H |
| [21 50 10 1] | Layer 1 | 0.3 | 10 | 1e4 | 170.4 | 7.32 | 0.78 | 7.60 | 0.79 | 8.66 | 0.72 |
| | Others | 0 | | | | | | | | | |
| [21 50 10 1] | Layer 1 | 0.3 | 10 | 1e3 | 48.36 | 6.66 | 0.76 | 7.29 | 0.77 | 6.62 | 0.73 |
| | Others | 0 | | | | | | | | | |
| [21 50 10 1] | Layer 1 | 0.3 | 10 | 1e2 | 20.09 | 7.00 | 0.75 | 7.18 | 0.76 | 6.02 | 0.68 |
| | Others | 0 | | | | | | | | | |
| [21 50 10 1] | Layer 1 | 0.3 | 100 | 1e3 | 73.19 | 6.58 | 0.77 | 7.88 | 0.79 | 7.65 | 0.73 |
| | Others | 0 | | | | | | | | | |
| [21 50 10 1]※ | Layer 1 | 0.3 | 100 | 1e2 | 8.40 | 6.62 | 0.78 | 6.70 | 0.80 | 6.63 | 0.69 |
| | Others | 0 | | | | | | | | | |
| [21 50 10 1] | Layer 1 | 0.01 | 100 | 1e2 | 6.64 | 6.42 | 0.77 | 6.97 | 0.81 | 6.86 | 0.68 |
| | Others | 0 | | | | | | | | | |
| [21 10 2 1] | Layer 1 | 0.01 | 100 | 1e2 | 7.47 | 7.20 | 0.78 | 7.43 | 0.78 | 8.15 | 0.72 |
| | Others | 0 | | | | | | | | | |
| [21 100 50 20 1] | Layer 1 | 0.5 | 100 | 1e2 | 4.77 | 8.07 | 0.74 | 9.87 | 0.79 | 9.63 | 0.63 |
| | Others | 0 | | | | | | | | | |
| [21 200 50 20 10 | Layer 1 | 0.5 | 100 | 1e2 | 2.73 | 13.1 | 0.71 | 14.8 | 0.78 | 16.0 | 0.63 |




| | | | | | | | | | | |
|---|---|---|---|---|---|---|---|---|---|---|
| 1] | Others | 0 | | | | | | | | |
| [21 200 100 50 20 10 5 1] | Layer 1 | 0.5 | 100 | 1e2 | 3.33 | 8.93 | 0.77 | 10.5 | 0.83 | 11.6 | 0.66 |
| | Others | 0 | | | | | | | | |
| [21 200 100 50 20 10 5 1] | Layer 1 | 0.5 | 100 | 50 | 2.53 | 12.6 | 0.79 | 13.7 | 0.85 | 16.5 | 0.67 |
| | Others | 0 | | | | | | | | |
| [21 50 40 30 20 10 5 1] | Layer 1 | 0.5 | 100 | 50 | 2.54 | 12.3 | 0.77 | 14.3 | 0.80 | 15.2 | 0.73 |
| | Others | 0 | | | | | | | | |

In this article ,we fixed the following four parameters in every situation: A-type of activation function(sigma); B-learning rate(1); C- zero masked fraction(0.5).

various parameters: D-non-sparsity penalty; E-number of epochs; F-batchsize.

G-RMS misfit of our result with other model; H-correlation coefficient of our result with other model.

※- selected sSAE neural network structure

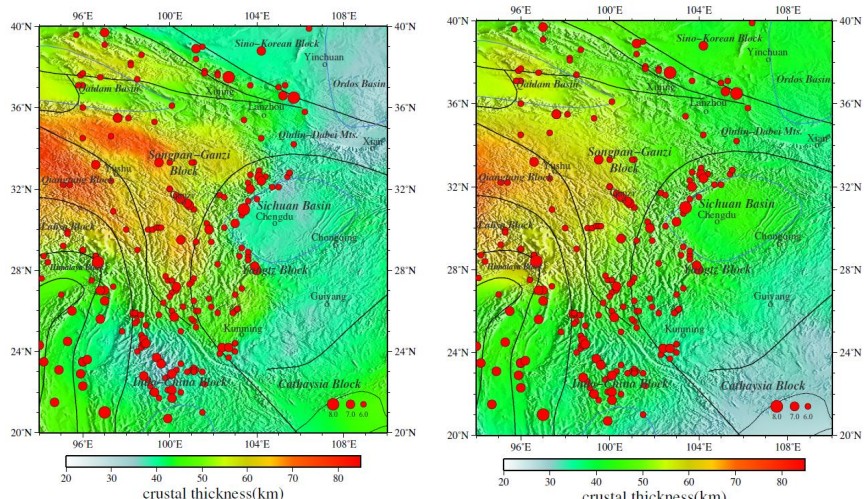

**Figure 3** crustal thickness of western Yangtze craton . The black lines in the figure show structure lines. The red dots show seismic events in this region from 1975  to 2015, and size of dot demonstrates size of magnitude from Ms6.0 to Ms8.0. These are  same to Figure4

**Figure 4** crustal thickness of model CUB2 from Shapiro&Ritzwoller (2002)

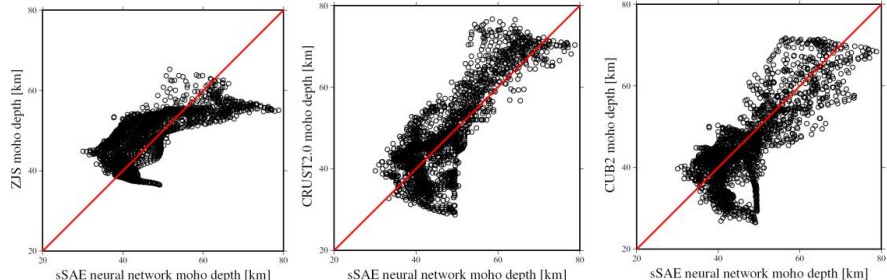

**Figure 5**  (From left to right) The correlation coefficient of our model with ZJS, CRUST2.0 and CUB2

## 4   Discussion

On the one hand, we can attain the moho depth and resultant geodynamic implication in research region from our result. We find our results are coincidence roughly with model ZJS,CUB2, CRUST2.0 (Fig.3,Fig.4, Fig.5), and the relatively good correlations of our result with CUB2,CRUST2.0 are shown in Fig.5. All have characteristics of  deep crustal thickness in the west of Longmen mountain and





relatively shallow in the east. Moreover, our results reveal more details: the eastern Tibetan Plateau crustal thickness is complex and changes largely with characteristic of deep west and shallow east. The average crust thickness is about above 60km, especially there is about 70-75km at Qiangtang block, under which there is a north dipping moho gradient zone. There is relatively shallow crust at Songpa-
Ganzi block and is characteristic of decreasing in northwest-southeast orientation. Model CUB2.0 tells us the crustal thickness of Sichuan basin is about 40km and is relatively smooth, however our model reveals there are some changes about crustal thickness in this region, that is crustal thickness is thin around Chengdu especially northeastward to Chengdu, in addition there is about 50km thick crust under Qinlin-Dabei fold belt, also we can get that crustal thickness of northeast to Sichuan basin is
about 45~48km.What's more, crustal thickness around Xi'an and Ordos basin is shallow about 35km. Conversely, change of crustal thickness in Sichuan-Yunnan block is sharp, which is 60km in northwest and 35km in southeast. All detailed information is consistence with Wang et.al(2010) who attained the crustal thicknesses estimated by the H-k stacking method based on the broadband tele-seismic data recorded at 132 seismic stations in Longmen mountains and adjacent regions( 26°~35°N, 98°~109°E ).
In addition, compared with the distribution of the epicenters during 1970-2015, great earthquakes in Sichuan and Yunnan have occurred in brittle upper crust, where moho depth changes sharply as to about 10km such as Longmen mountain fault zone where occurred great Ms 8.0 Wenchuan earthquake in 2008 and Ms 7.0 Lushan earthquake in 2013. The reason may be that main fault cut moho where material in crust and mantle exchange and accumulating press induce a series of earthquakes
frequently.

On the other hand, our results show deep learning neural networks can invert crustal thickness effectively due to their owning capability to represent complex functions:

Test errors of deep learning neural network may be influenced by the number of layer in networks which shows more layers induce smaller test errors, which we can attain from Table 1 when the
number of layer in networks adds from three to six, test error decreases from 1.7e-4 to 2.5e-6. In addition, training parameters as batchsize decrease from 1e4 to 1e3 and test error decreases from 1.7e-4 to 2.5e-5. Also when epochs increase from 10 to 100, corresponding test error decreases from 2.0e-5 to 8.4e-6.

The robustness of deep learning neural networks is strong. When the number of layers in network
achieves six, changes of the number of neurons in each layer have little influence on test errors which is about 2.5e-6.

The neural network structure shown in ※ from table 1 reveals misfits of our model with model CUB2, CRUST2.0 ZJS are relatively low with 6.62, 6.70 and 6.63, and corresponding correlation coefficients are relatively high with 0.78, 0.80 and 0.69 respectively, however, test errors is 8.4e-6 and
is not minimum. This tells us test error may be not the only criterion determining which neural network is best because small test error may be induced by overfit.

## 5    Conclusion and remarks

Taking use of sSAE deep learning network, we present moho depth map of eastern Tibet and western Yangtze craton (Fig.3). The data sets consist of phase velocities of Rayleigh waves from
Xie(2013) at discrete frequency of 10.0, 12.5, 15.0, 17.5, 20.0, 22.5, 25.0, 27.5, 30.0, 32.5, 35.0 mHz and derived group velocities of Rayleigh waves at discrete frequency of 10.0, 12.5, 15.0, 17.5, 20.0, 22.5, 25.0, 27.5, 30.0 mHz. We conclude that:

(1) For all our simulations we use sSAE with different neural network structures which are decided by many factors such as the number of layers and neurons in neural networks, optional
parameters as the number of epoch and batchsize, type of activation function, values of learning rate and non-sparsity penalty and so on. We find that the number of hidden units is not a crucial parameter and networks with different number of hidden units give similar results, however batchsize is an important factor for results.

(2) After invert these twelve networks, different networks produced different results. Compared
with other crustal thickness models we find network with the smallest test error is not the best result always. When test errors achieve some value, the misfits are high and correlation coefficients are low, which we think it is maybe caused by overfit. In our future work, we'll focus on how to resolve this problem in using sSAE.

(3) We present a crustal thickness model for eastern Tibet and western Yangtze craton. Compared
our model with current knowledge about crustal structure as represented by ZJS,CRUST2.0, CUB2. The overall agreement with these three models is very good, and agreement is generally better with CUB2.





(4) The results are obtained using a neural network approach sSAE which is widely and successfully used in pattern recognition . As we all know, geophysics is so complex that we should analysis and enhance neural network to apply to these complicated problems.

*Acknowledgements.* The authors are grateful to Xie for providing the phase velocity maps, and Zhu J S for making the model available. Our work are funded under Active Fault Study Team of Chengdu University of Technology, grant number 10912-KYTD201505.

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
