# Peer review of "Inverting Rayleigh surface wave velocities for eastern Tibet and western Yangtze craton crustal thickness based on deep learning neural networks"

_Nonlinear Processes in Geophysics, 2016_

## Referee Comment (RC1) · Anonymous Referee #1 · 21 Oct 2016

Cheng, Lui & Li inverted surface-wave phase-velocity maps from ambient noise to obtain crustal thickness for eastern Tibet and western Yangtze Craton. They applied a three steps procedure: in a first step, they collected the phase velocity maps and extrapolated the phase velocities into group velocities. In a second step, they performed a joint inversion of phase and group velocities using neural network techniques in order to obtain crustal thickness of western China. Finally, a comparison between their results and other published models is performed.

**Novelty**
This paper presents new results that are in good agreement with existing models and published results.

**Fluency and precision of the text**
Check the English please. Some parts of the manuscript are difficult to understand. A few examples: Page 1, lines 27-28 "As we all know, the more we know the characteristic and composition of crust which is an important part of lithosphere, the further we investigate deep earth".

Page 1, lines 36 What does "adjants" stands for?

Page 1, lines 40-42 "in regions with good data coverage and uncomplicated structure but in regions with poor or no data coverage or complicated structure crustal thickness estimates are largely extrapolated"

Those statements (and several others in the manuscript) are not clear and would need some rephrasing. In addition, several typos and missing capitalization of letters in nouns should be checked carefully. Several sections of the text should be checked carefully. The consistency of the text is not always straightforward.

**Bibliography**
In this paper, a strong effort was made to provide relevant references to the methodology. However, several aspects would require a more extensive referencing.

In the last decade, and the deployment of hundreds of seismic stations in mainland China, several important publications arose, and several focusing on crustal structure have been omitted in this manuscript:

Legendre, C. P., Deschamps, F., Zhao, L., & Chen, Q. F. (2015). Rayleigh-wave dispersion reveals crust-mantle decoupling beneath eastern Tibet. Scientific reports, 5.

Sun, X., X. Song, S. Zheng, Y. Yang, and M. Ritzwoller (2010), Three dimensional shear wave velocity structure of the crust and upper mantle beneath China from ambient noise surface wave tomography, Earthquake Sci., 23, 449–463.

Xu, L., Rondenay, S. & Van Der Hilst, R. D. (2007), Structure of the Crust beneath the southeastern Tibetan Plateau from teleseismic Receiver Functions. Physics of the Earth and Planetary Interiors. 165, 176–193.

Yao, H., R.D. Van Der Hilst, and M. V. De Hoop (2006), Surface-wave array tomography in SE Tibet from ambient seismic noise and two-station analysis–I. Phase-velocity maps, Geophys. J. Int., 166(2), 732–744.

Zhang, Q., E. Sandvol, J. Ni, Y. Yang, and Y. J. Chen (2011), Rayleigh wave tomography of the northeastern margin of the Tibetan Plateau, Earth Planet. Sci. Lett., 304(1-2), 103–112.

Zheng, S., X. Sun, X. Song, Y. Yang, and M. H. Ritzwoller (2008), Surface wave tomography of China from ambient seismic noise correlation, Geochem. Geophys. Geosyst., 9, Q05020, doi:10.1029/2008GC001981.

Zhou, L., J. Xie, W. Shen, Y. Zheng, Y. Yang, H. Shi, and M. H. Ritzwoller (2012), The structure of the crust and uppermost mantle beneath South China from ambient noise and earthquake tomography, Geophys. J. Int., 189(3), 1565–1583.

Also, a recent review of inversion strategies provide many discussions that would be emphasized in the manuscript:

Lebedev, S., Adam, J. M. C., & Meier, T. (2013). Mapping the Moho with seismic surface waves: a review, resolution analysis, and recommended inversion strategies. Tectonophysics, 609, 377–394.

In addition, several references contain some typos, mostly missing capitalization of some names (as "rayleigh" instead of "Rayleigh"), and some errors in the citation occur

[(Ueli Meier et al. (2007), (Zhu J S et al., 2012) or (Liu, 2015)].

Another point is the lack of consistency in the referencing style, as well as the abbreviations used for some journals. For example, Montagner et al. (1988) has been published in Geophysical Journal International, and its short form is "Geophys. J. Int.", not "Geophys. J.".

The Mineos package (page 4, line 28) is not referenced (page 4, line 31) at its first occurrence. Please check the references carefully.

**Comments on the figures**
Table 1 provides many parameters but no unit is given.

Regarding the figures, I would suggest to merge Figures 1 and 2. Both describes the auto-encoder with one or two hidden layers, and could easily be merged.

Similarly, Figures 3 and 4 could be merged. They both show crustal thickness from this study and from another model, used later for comparison.

In the text, several places are mentioned but are not located on any map, as the Wenchuan or Lushan earthquakes, the Longmenshan region, ... In addition, in Figures 3 and 4, the caption doesn't mention that the blue lines are the boundaries of sedimentary basins.

**Technical comments**
First, the authors wrote (page 4, line 37) that the phase and group velocities of surface-waves are not sensitive to similar depth layers. However, they extrapolates the phase velocities into group velocities, using the formula (4) , line 1-2, page 5. But the resulting periods for group and phase velocity dispersion curves are similar (10.0 - 30.0 mHz for group-velocity and 10.0 - 35.0 mHz for phase-velocity). This is not consistent with

the period range described page 5, line 26, with Rayleigh phase velocities and group velocities (10 - 37.5 mhz). (Note that it should also be mHz and not mhz).

Basically, the authors only have constraints on Rayleigh-wave phase velocity (from Xie et al. 2013), that are derived for approximated group velocities, and joint inverted for crustal thickness. Why not using the Love-wave models in order to add different constraints to their inversion?

The authors used the phase-velocity model (from Xie et al. 2013) for periods between 33 and 100 s (10 - 30.0 mhz). But Rayleigh-wave phase velocities at periods of 33 s are mostly sensitive to depths of 30-80 km. In some regions (Sichuan Basin), the authors found some Moho depths shallower than 30 km. Are those depths realistic?

Why don't the authors used periods of 8-40 s (25-125 mHz) as in Xie et al., (2013) to have additional constraints on the regions with shallow crust?

In the same way, Xie et al., (2013) used a grid of 0.5*0.5 °. Why do the authors down-sample those maps to 2*2 ° (page 2, line 16)?

**Comments on the method**

Another point that the author did not mention is how they invert the surface-waves velocities (phase and group) for Moho depth. They only mention (page 7, lines 21-25) that the dispersion curves are inverted for crustal thickness using 3 to 6 layers. Some additional information of the methodology seems needed. Did they used 1D, 2D or 3D sensitivity kernels? How did they defined the Moho discontinuity (velocity contrast, specific velocity, ...)?

The authors mention (page 7, line 35-36) "test error may be not the only criterion determining which neural network is best". So what are the other criteria that needed to be taken into account?

---

## Author Comment (AC1) · 2 Nov 2016

We thank anonymous referee for his working for this paper, who has given many good suggestions, which we are incorporated in this revised work. We answer all questions in attached file named "npg-2016-39_Author Reply.pdf". Also we upload the revised file and revised marked-up file named "npg-2016-39-revised file(Referee 1).pdf" and "npg-2016-39-revised marked-up file(Referee 1).pdf" respectively. All these three files compress into a file named npg-2016-39.zip.

[Figure]

Please also note the supplement to this comment:
http://www.nonlin-processes-geophys-discuss.net/npg-2016-39/npg-2016-39-AC1-supplement.zip

---

## Referee Comment (RC2) · C. Nunn (Referee) · 11 Nov 2016

**Review of : Inverting Rayleigh surface wave velocities for eastern Tibet and western Yangtze craton crustal thickness based on deep learning neural networks**

Xian-Qiong Cheng[1] , Qi-He Liu[2] , Ping-Ping Li[1]

I enjoyed reading the paper, and thought it was a very interesting new application of a technique. Neural networks is not something I know much about, but I thought it was interesting that the authors applied this non-linear technique to the non-linear problem of crustal thickness. In my opinion, it is definitely something that should be published in this journal.

However, there are several points which should be addressed:

Major:

1) I did not understand how the authors built their model of crustal thickness from the data. For example, did they build some layers, and add them up for crustal thickness. What information allowed them to decide they were in the crust or mantle? This was not discussed at all. This discussion needs a figure.

2) p4, 27-29  I wondered whether the choice of PREM as a starting point for training the models was a good choice. PREM is very different from much of the Tibetan plateau and surrounding areas. On the other hand, 100,000 synthetic models sounds pretty impressive! I think this section needs a figure showing the models as a depth profile (either just each model plotted on top of each other - or some kind of probabilistic model). Then the authors should either justify their use of PREM as a good choice - or include some other models which take into account larger crustal thickness.

3) I was not clear during the paper whether crustal thickness really meant thickness, or whether it was depth below sea level. Since the plateau is at 5 km above sea level, this is quite important.

4) Topographic effects. I think the authors should mention whether they are think there are errors associated with the surface topography - and if anything can be done about these errors.

Minor points:

Title: I recommend:
Inverting Rayleigh surface wave velocities for crustal thickness in eastern Tibet and the western

Yangtze craton based on deep learning neural networks

p1, abstract 'Based on test errors 15 and misfits with other crustal thickness models, we select the optimized one as crustal thickness for study areas. ' There is no obvious reason to assume that the other models are any better than yours! So I wasn't sure why you would want to choose your favourite model based on a comparison (This comment doesn't apply to the later section where you compare your models with other models - I thought that was interesting).

p1, 31 The Moho (Mohorovičić discontinuity) is a seismic discontinuity, and may not even be present. It is not the same as crustal thickness (although it has a strong correspondence with it).

p1, 33 'has significant effects on fundamental model surface wave' - you could reference other papers as well here

p1, 36 adjants => surrounding areas

p1, 42 defaults => defects

p1, 48-50 - be careful with the wording. For example Shapiro and Ritzwoller 2002 is shear-wave velocity model - not directly about crustal thickness.

p2, 18, traditional shallow neural networks => please explain what these are first

p2, 29 deep learning neural networks. Moreover, our deep learning neural networks train on vast synthetic models.  => please explain

p2, 31, 'Lastly, our results show changes of the number of neurons in each layer have little influence on test errors when the numbers of network layer achieve six and test errors are about 2.5e-6 ' => this didn't really belong here, before you have explained about layers

p2, 57 'shallow neural networks'  => this still hasn't been explained

p3, 12, Add references

p4, 28, PREM - please explain what PREM  and Mineos are, and reference them.

p4, 39 'Since a larger part of the signal is affected by the crustal structure, combination two types of data will constrain crustal thickness better in the presence of noise.' Reference this statement if you can

p5, 3 Based on Rayleigh wave
phase velocity from ambient noise(Xie et.al,2013), we compute corresponding group velocity

This sentence didn't make sense the first time I read it - I didn't understand that you were using *data* from Xie 2013)

p5, 18 After trying many times, we find the proportion of training data set to test one is 3:1 is reasonable. A figure would be helpful here.

p5, Please explain this table further. Which is a good result and why, and give a bit more explanation about the table headings.

p5, 28 as shown in table 1 shown in ※. What is this symbol?? It is referred to several times.

p5, 25-35 This table mixes method, results and discussion. It would be better to separate them..

Fig 3 - Indicate that the left side is your work

p5 - Add a new figure showing some velocity profiles across the region - as explained above I cannot see how you have arrived at Fig 3.

p7, 49-53 - Interesting point, but explain a bit more

General:

English - There are some problems with the English text which make the paper quite difficult to read. A good revision of the English would be helpful.

I found it difficult to follow your explanation of the development/background of neural networks. More logical explanation - and not introducing terms before they are discussed would be good. (The description of your own method and its application was good and I was able to follow that).

I hope you find these comments useful.

Ceri Nunn

Ludwig-Maximilians-Univesität, Munich

p

---

## Author Comment (AC3) · 29 Nov 2016

We thank Ceri Nunn for her working for this paper, who has given many good suggestions, which we are incorporated in this revised work. We answer all questions in attached file named "npg-2016-39_Author Reply(Referee 2).pdf". Also we upload the revised file and revised marked-up file named "npg-2016-39-revised file(Referee 2).pdf" and "npg-2016-39-revised marked-up file(Referee 2).pdf" respectively. All these three files compress into a file named npg-2016-39(Referee 2).zip.

[Figure]

Please also note the supplement to this comment:
http://www.nonlin-processes-geophys-discuss.net/npg-2016-39/npg-2016-39-AC3-supplement.zip